# Estimating the 18-Year Threshold with Third Molars Radiographs in the Southern Italy Population: Accuracy and Reproducibility of Demirjian Method

**DOI:** 10.3390/ijerph191610454

**Published:** 2022-08-22

**Authors:** Mario Caggiano, Giuseppe Scelza, Alessandra Amato, Raffaele Orefice, Simona Belli, Stefano Pagano, Chiara Valenti, Stefano Martina

**Affiliations:** 1Department of Medicine, Surgery and Dentistry “Scuola Medica Salernitana”, University of Salerno, Via Allende, 84081 Baronissi, Italy; 2ASL Napoli 1 Centro, Strada Comunale del Principe, 13/a, 80145 Napoli, Italy; 3Odontostomatological University Centre, Department of Medicine and Surgery, University of Perugia, 06132 Perugia, Italy

**Keywords:** third molars, dental imaging, age estimation, Demirjian method

## Abstract

The estimation of the age of the majority of living subjects is widely required nowadays due to the presence of unidentifiable individuals, without documents and general information, involved in migration or legal procedures. Dental age estimation (DAE) is a valid method for investigating the age of subjects. The aim of this study was to evaluate the accuracy of the Demirjian method in a limited age group (16–24 years) in differentiating between older and younger than 18 years. From an initial sample of 17,594 radiographs, 460 were selected meeting the inclusion criteria. Two dentists provided the age estimate according to the Demirjian method, with a simplified approach based on the development of the third molars. The presence of a developmental stage of H for at least one third molar allowed to establish the major age if the other third molars, inferior or superior, have reached a stage equal or superior to F, with an accuracy of 90.2% and a predictive positive value of 91.6%. Thirty-three patients showed the development of at least one third molar (Stage H) before the age of 18 years while six patients showed the development of all four third molars with root completion (stage H) before the age of 18 years. When all third molars reached stage H an individual was over 18 years old in 97.4% of cases. In presence of one third molar on stage H and a stage equal or superior to F for the other third molars the probability of being of major age was 91.6%.

## 1. Introduction

In recent years, age estimation on living subjects has been more and more requested, due to the increasing number of unidentifiable individuals, because of lack of documents, and those involved in migration or legal proceedings [1]. The main objective is to recognize whether a subject has reached the age of majority, which in Europe, as in many other countries of the world, is set at the age of eighteen years. Indeed, if the biological differences between an adolescent subject of 17 years and one of 19 years are negligible, from the forensic point of view it is fundamental to establish the status of a minor individual, for the different legal implications. This field of investigation has attracted so much interest that it has been necessary to standardize methodologies for age estimation in living persons that include analysis of physical, bone, and dental development. These investigations are conducted with the aim of matching sexual, skeletal, and dental maturity to the age of the subject under study [2]. Age estimation can be carried out through methods that study the somatic and sexual characteristics of the individual (e.g., secondary sex characteristics, predicted percentage of mature stature) [3]. The physical examination, evaluating some somatic traits (weight, height, overall body development) and the degree of sexual maturation can provide relevant information, especially in pre-pubertal age, and therefore not applicable in the determination of the major age [4]. However, the most reliable methods involve bone age, which indicates a person’s skeletal maturity and can be useful in estimating chronological age more accurately [5]. Different methods to calculate age according to X-ray findings have been widely adopted, i.e., wrist-hand X-ray, hip X-ray, cervical spine X-ray, CT of the clavicle, and orthopantomography [2].

The gold standard for the assessment of bone growth is the hand and wrist X-ray [6]. Until the end of adolescence, when bone elongation has ended, the bones of the hand and wrist can be good indicators of an individual’s age. X-rays of the hand and wrist allow age estimation by comparison with standard values of maturation and bone age [5]. Wrist-hand X-ray offers the best compromise between diagnostic accuracy and radiation exposure according to the ALARA principle, a safety recommendation designed to minimize radiation doses “as low as reasonably achievable”. The effective radiation dose is near 0.001 mSv, so it is the most adopted approach in the pediatric population; however, this method is not applicable by the age of 18 years [7], because a complete development of the anatomical hand-wrist district is observed on average at 16.5 years of age. Consequently, a subject could have a complete ossification but not have reached 18 years of age [8]. As ossification and sexual maturation are complete by the age of 18, in individuals aged 18–22 years a reliable method is CT for ossification of the medial end of the clavicle. However, this approach exposes to high radiation doses and cannot be applied as a screening method to establish age in this group of population in terms of clinical, safety, and economic considerations [2,9]. A valid alternative for determining bone age is dental age estimation (DAE), which includes techniques such as gingival emergence, eruption sequence of the teeth, or, more frequently, radiographic assessment of the degree of third molar mineralization [10]. In addition, orthopantomography shows the best risk-benefit and cost effectiveness ratio, with an effective radiation dose near 0.005 mSv compared to 3–4 mSv of a CT scan. Several approaches have been suggested in dental age analysis. In 1955 Gleiser and Hunt [11] noted that the different phases of odontogenesis, which can be analyzed on radiographic investigations on the basis of the progressive calcification of the hard tissues of the tooth, had more relevance than the eruptive sequence in determining the age of the patient examined. The authors, in their longitudinal observational study, arbitrarily chose the lower permanent first molar and evaluated its development based on radiographic examinations performed every six months. As a result, 15 stages of tooth development were differentiated and correlated with specific patient ages. More recent studies have applied the method of Gleiser and Hunt (with a modification by Kohler) for estimating the age of the majority, showing an accuracy of about 90% [12]. Age evaluation becomes more complex when the roots of the mandibular second molar complete their development as the apical foramen matures. In the period between mid-adolescence and 20 years of age, the third molar represents the only dental indicator that can be used to define the age of the majority because it still has margins of development [13]. Over the years, several classification methods have been devised to assess the stages of mineralization of teeth [14,15,16,17,18,19]. In particular, Kullman’s method [17] analyzed the radiological development of the third molar root. However, the authors recorded low reproducibility between operators, low precision, and a standard deviation of approximately one to two years around the mean age of the different developmental stages. Furthermore, the method of Nolla et al. [18] made it difficult to identify the corresponding stage of maturity on the dental radiographs due to the low-quality graphic representations. Another method for assessing chronological age based on the relationship between age and measurement of the open apices in teeth was the Cameriere method, which was shown to be accurate, but tended to underestimate the age of the subjects [19]. The Demirjian method represents by far the most employed for age assessment, due to its simplicity as it clearly defines the stages of third molars development, decreasing the inter- and intra-observer discrepancy [7,20].

This age analysis method is based on the evaluation of panoramic X-rays on which each lower third molar is rated based on an eight-stage scale from A to H according to development data [21]. In the literature, there are many studies on this method to assess the 18-year-old threshold in other populations [22,23,24,25]. However, there are no studies conducted in Italy on this method on a large sample of subjects close to the age of the majority. The aim of this study was to evaluate the accuracy of a modified Demirjian method on a limited age group (16–24 years) in differentiating between older and younger than 18 years. 

## 2. Materials and Methods

This retrospective study involved the analysis of 17,594 orthopantomographies performed in the same dental imaging center in Naples, Campania, Italy.

The panoramic X-rays considered in the study had the following inclusion criteria:Availability of patient’s birth data and date of exposure to X-ray examination;Patients aged between 16 and 24 years.Orthopantomograms with the following criteria were excluded:Absence of more than one third molar;X-rays with unclear images;Third molar pathologies;Presence of any syndromes;

The final sample that met inclusion criteria consisted of 460 patients in the age range of 16–24 years (88 subjects were under 18 years old, 372 were over 18 years old). Of these, 225 were males (mean age = 20.12 ± 2.28), and 235 females (mean age = 20.28 ± 2.35). 

Two dentists trained in dental imaging with more than 10 years of experience (SM and RO) were asked to provide age estimation according to the Demirjian method, based on the third molar development [21]. The original method was simply based on the presence of at least one inferior third molar at stage H; if the analysis was positive, the patient was considered of major age. With our modified approach a patient was considered of major age even in presence of a third inferior molar at stage G with a third superior molar at stage H. All evaluations were blindly performed. Each observer was invited to perform two sessions of evaluation with a 3-week time interval (T = initial; T2 = 3 weeks). All patients gave informed consent to perform the examination, but no study-specific data consent was necessary because the examiners did not know the confidential data of the patients, only their date of birth and X-ray exposure.

Frequencies for categorical data were computed. A chi-square test was used to assess the association between sex (male vs. female) and a Wilcoxon signed-rank test to compare upper and lower molar sides (right vs. left). A weighted Cohen kappa (κ) test was used to assess the interobserver and intraobserver agreement. The range of variation of the weighted κ statistic is between 0 for no agreement and 1 for perfect agreement with five intermediate levels: slight agreement (0.01–0.20), fair agreement (0.21–0.40), moderate agreement (0.41–0.60), substantial agreement (0.61–0.80), and almost perfect agreement (0.81–0.99) [26]. The specificity, sensitivity, and accuracy of the method with the positive and negative predictive values were calculated. The accuracy is the overall probability that a patient is correctly classified and was calculated in this way: Sensitivity × Prevalence + Specificity × (1 − Prevalence). Confidence intervals for sensitivity, specificity, and accuracy are “exact” Clopper-Pearson confidence intervals, while the predictive values are the standard logit confidence intervals given by Mercaldo et al., 2007 [27]. A standard statistical software package (SPSS, version 27.0; SPSS IBM, Armonk, NY, USA) was used. The level of significance was set at *p* < 0.05.

## 3. Results

The third molar formation was examined in 460 patients of both sexes and no significant differences were found in third molar development between males and females (χ^2^ = 2.76, *p* = 0.097) and between left and right molars of the same arch (Upper Molars *p* = 0.68; Lower Molars *p* = 0.29). Intraobserver agreement was almost perfect for both observers (SM κ = 0.83, RO κ = 0.86). Interobserver agreement was substantial with a weighted κ of 0.76. Considering only the evaluation of the lower molar the interobserver agreement was considered almost perfect with a weighted κ of 0.84 (Table 1).

Thirty-three patients (mean value between the four examinations) showed the development of at least one third molar (Stage H) before the age of 18 years. The presence of a developmental stage of H for at least one third molar allowed to establish the major age if the other third molars, inferior or superior, have reached a stage equal or superior to F, with an accuracy of 90.2%, a predictive positive value of 91.6% and a predictive negative value of 81.8%. Nevertheless, only six patients (three males and three females with a mean age of 17.6 ± 0.2) showed the development of all four third molars with root completion (stage H) before the age of 18 years. Considering the evaluation of stage H in all four third molars the predictive positive value increased to 97.4%. All the data of accuracy, sensitivity, specificity, positive predictive value, and negative predictive value are resumed in Table 2.

## 4. Discussion

We propose a simplified approach based on the Demirjian method that seems to be reliable as, according to our population data, the presence of a score of H for at least one third molar is enough to establish the major age if the other third molars, inferior or superior, have reached a stage equal or superior to F. The advantages of the Demirijian method that emerged from our study are the high intra- and inter-operator reproducibility, the accuracy of the results, and the positive predictive value. Indeed, this method may provide a useful tool for age estimation for medico-legal purposes as those daily requested nowadays for asylum seekers and refugees or including criminal identification and legal responsibility, and for other social events [24].

Regarding the reproducibility, the k-values for intra-observer (0.88) and inter-observer (0.84) agreement found in our study were lower than the paper on the same method by Mohammed et al. [24], who recorded 0.93 and 0.92 respectively. This difference could be due to the age range of the sample, which was greater in the mentioned study (9–21 years old). In fact, in younger patients, it may be easier to assess accurately the stages of formation of the third molar. High reproducibility is one of the main reasons for preferring this method to the simple observation of the eruptive sequence of the teeth, which is unreliable and not useful for the goal of determining the adult age [28]. This is because the assessment of third molar eruption is not reliable due to the absence of a precise age of eruption and the high presence of inclusions and agenesis [29]. A study by Gambier et al. reported a high probability that when the third molars have erupted in the arch, the subject is over 18 years old, but the authors themselves stated that the assessment of the eruption of M3s alone was not sufficient for the determination of whether or not an individual is aged 18 years or older [30].

By analyzing the sample, we found no significant differences between men and women, similar to the study by Mohammed et al. [24] and in contrast to other studies that reported statistically significant differences [23,25,31]. We also found no differences between right and left molars, according to the results of previous studies [23,25,31,32].

In our study, a total of 33 subjects (7.2%) showed the development of at least one third molar (Stage H) before the age of 18 years. These data were higher than those of a previous study [11] that reported 4.3% of individuals with one third molar in Stage H before the age of 18 years. These discrepancies may be due to different ethnicities, the size of the sample, and the expertise of the examiners. Anyway, these findings were consistent with previous studies that showed a possible age overestimation of the Demirjian method [33,34,35]. According to Quaremba et al., this method remains the recommended way to assess individual dental maturity, but it should definitely be considered unsuitable for application, particularly as regards the most disputed age range 14–16 years [35,36]. Nevertheless, the probability of an individual being older than 18 years old in case of the presence of four wisdom teeth in stage H was 97.4% in our study. This value is in agreement with the results of the study by Lewis et al. [20] which found a chance of 94.12% for males and 100% for females. Based on these findings, we could consider the overestimation of age in the Demirjian method to be more important in the early stages than in the stage H, confirming the usefulness of the method in assessing the age of the majority. 

Moreover, the sensitivity value (96.7%) of the method found in this study was consistent with the results of the study by Kanchan et al. [23] reporting values of 94.7% in males and 97.1% in females.

Some studies indicated that the third molar had the greatest variation in morphology, size, and time of development and eruption compared with the other teeth. Thus, the assessment of its developmental stages and relation to chronological age might not be reliable [32,37,38]. However, the results of our study were in agreement with other findings demonstrating that assessment of the third molar developmental stages was a reliable and useful method for major age estimation [39,40]. In addition, Digital Panoramic X-ray Machine Systems are of easy installation and minimum staff training, so it could be possible to estimate age directly in the welcome refugee point with a simple method, cost, and time effective, also in teleradiology, with a minimum exposure dose [41].

The data described above may provide a reference for the forensic application of the third molar examinations to the population. Additional studies with a larger study population extending the sample to the whole of Italy should be conducted to minimize the differences that may exist between different geographic areas within the same country. However, the geographical/ethnic differences seem to be small, making the method useful irrespective of the ethnic profile of the subjects [42]. Nonetheless, it is important to remember that age estimation concerns biology and variation is to be expected. One must be cautious in the interpretation and application of results obtained as the methods used help to determine an individual’s overall maturity and may only approximate the chronological age. This further stresses the need for multifactorial methods of age estimation (physical examination, bone, and dental development) which when used according to their reliability may help to control the variation that occurs with age when a single indicator is used. 

## 5. Conclusions

From this study of 460 patients, it can be surmised that the probability of an individual being older than 18 years is 97.4% when all the third molars have attained stage H. In presence of one third molar on stage H and a stage equal or superior to F for the other third molars the probability of being of major age is 91.6% with an almost perfect intraobserver agreement and a substantial interobserver agreement.

## Figures and Tables

**Table 1 ijerph-19-10454-t001:** Cohen weighted κ coefficient for intraobserver and interobserver agreement at two time intervals (T1 and T2).

	Κ	*p*	Interpretation
SM T_1_ vs. SM T_2_	0.83	<0.001	Almost perfect
RO T_1_ vs. RO T_2_	0.86	<0.001	Almost perfect
SM (T_1_ + T_2_) vs. RO (T_1_ + T_2_)	0.76	<0.001	Substantial
SM T_1_ low ° vs. SM T_2_ low	0.86	<0.001	Almost perfect
RO T_1_ low vs. RO T_2_ low	0.90	<0.001	Almost perfect
SM (T_1_ + T_2_) low vs. RO (T_1_ + T_2_) low	0.84	<0.001	Almost perfect

° indicates the evaluation of lower third molars.

**Table 2 ijerph-19-10454-t002:** Accuracy, sensitivity, specificity, positive predictive value and negative predictive value considering two different conditions: development of at least one third molar (Stage H) or development of all four third molars.

	At Least 1 Molar in Stage H	95% CI	4 Molars in Stage H	95% CI
**Accuracy**	90.2%	88.7% to 91.5%	67.6%	63.1% to 71.9%
**Sensitivity**	96.7%	95.7% to 97.6%	61.6%	56.4% to 66.5%
**Specificity**	62.5%	57.2% to 67.6%	93.2%	85.8% to 97.5%
**Positive predictive value**	91.6%	90.5% to 92.6%	97.4%	94.6% to 98.8%
**Negative predictive value**	81.8%	77.1% to 85.7%	36.4%	33.3% to 39.8%

## Data Availability

Not applicable.

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
