# Peer review of "Estimating the 18-Year Threshold with Third Molars Radiographs in the Southern Italy Population: Accuracy and Reproducibility of Demirjian Method"

_ijerph, 2022, doi:10.3390/ijerph191610454_

Round 1

Reviewer 1 Report

This manuscript aims to evaluate the accuracy of the Demirjian method in discriminating between major and minor. The authors present valuable results that could be useful in forensic science. Biological age plays a critical role in estimating chronological age. Knowing a person's age is important for criminal law, medicine, and forensics. It is especially important to eliminate the factor of overestimation of estimated age.

I would like to encourage the authors in their valuable work. However, the article to be published contains some recommendations of mine that should be considered. Both minor and major suggestions should be carefully reviewed. Please see the comments below.

- Revise the title, e.g., "Estimating the 18-year threshold with third molar radiographs in the southern Italian population (accuracy and reproducibility of the Demirjian method)."

- The description of the sample size and the OPGs selected for evaluation that meet the inclusion criteria is mentioned in the Results section, but I suggest that it be included in the Materials and Methods section.

- When you talk about exclusion criteria or study limitations, I suggest mentioning impacted and extracted teeth - how did you evaluate them?

- In the statistical analysis, the authors should mention the normality distribution of the data. Did you calculate the synchrony between the arches, or in words: the bilateral and collateral differences? It is important to know if we can replace the left mandibular third molar with the right mandibular third molar (the same is true for the maxilla). In the case of collateral synchrony, it should be obvious that the differences should be significant due to the different development of the mandibular and maxillary teeth. Did you observe this in your study as well?

- In the Material and Method section you mention, "The specificity and sensitivity of the method with the positive and negative predictive values were calculated" - how exactly did you proceed? Did you calculate the accuracy of the dental age estimate based on the differences between dental age and chronological age? Or did you perform a receiver operating curve analysis (ROC) to determine the Area Under the Curve (AUC) values of the method that distinguishes adults from minors? I strongly recommend that this be revised and discussed in more detail, as the accuracy of the method is part of your title and the goal of the study. Based on the above, I also recommend using, for example, boxplots or other graphical representations to illustrate accuracy with the respect to sex.

- In the discussion section, I suggest highlighting the fact of overestimation of the Demirijan method, and are there any suggestions for improvement?

- In the Discussion section, you state, "However, the eruption sequence is unreliable and not useful for the goal of determining adult age for several reasons." I would disagree, as the 2019 study by Gambier et al (Contribution of third molar eruption to the estimation of the forensic age of living individuals) comes to different conclusions. Please include them in your discussion.

Author Response

REVIEWER 1

This manuscript aims to evaluate the accuracy of the Demirjian method in discriminating between major and minor. The authors present valuable results that could be useful in forensic science. Biological age plays a critical role in estimating chronological age. Knowing a person's age is important for criminal law, medicine, and forensics. It is especially important to eliminate the factor of overestimation of estimated age.

I would like to encourage the authors in their valuable work. However, the article to be published contains some recommendations of mine that should be considered. Both minor and major suggestions should be carefully reviewed. Please see the comments below.

Thank you very much for your appreciation and for the comments and suggestions.

- Revise the title, e.g., "Estimating the 18-year threshold with third molar radiographs in the southern Italian population (accuracy and reproducibility of the Demirjian method)."

We changed the title according to the suggestion.

- The description of the sample size and the OPGs selected for evaluation that meet the inclusion criteria is mentioned in the Results section, but I suggest that it be included in the Materials and Methods section.

We moved the description of the sample size in the Materials and Methods section as suggested.

- When you talk about exclusion criteria or study limitations, I suggest mentioning impacted and extracted teeth - how did you evaluate them?

It is not important to know whether the missing third molar has been extracted or is agenetic for our assessment; for these reasons, we changed the exclusion criteria to "Absence of more than one third molar”.

- In the statistical analysis, the authors should mention the normality distribution of the data. Did you calculate the synchrony between the arches, or in words: the bilateral and collateral differences? It is important to know if we can replace the left mandibular third molar with the right mandibular third molar (the same is true for the maxilla). In the case of collateral synchrony, it should be obvious that the differences should be significant due to the different development of the mandibular and maxillary teeth. Did you observe this in your study as well?

We did not assess the normality distribution because the data are all dichotomous. A Wilcoxon signed-rank test was performed to investigate the differences between right and left side. We added this information in the Materials and methods section and in the Results section.

- In the Material and Method section you mention, "The specificity and sensitivity of the method with the positive and negative predictive values were calculated" - how exactly did you proceed? Did you calculate the accuracy of the dental age estimate based on the differences between dental age and chronological age? Or did you perform a receiver operating curve analysis (ROC) to determine the Area Under the Curve (AUC) values of the method that distinguishes adults from minors? I strongly recommend that this be revised and discussed in more detail, as the accuracy of the method is part of your title and the goal of the study. Based on the above, I also recommend using, for example, boxplots or other graphical representations to illustrate accuracy with the respect to sex.

In the Materials and method section, we tried to better define how we calculated accuracy and in the Results section we added another table with accuracy, sensitivity, specificity, positive and negative predictive values. We also added the chi-square test results between male and female and there is no statistically significant differences, which is why it would not have actually made much sense to make different boxplots among males and females.

- In the discussion section, I suggest highlighting the fact of overestimation of the Demirijan method, and are there any suggestions for improvement?

We tried to extend this paragraph according to your suggestion.

- In the Discussion section, you state, "However, the eruption sequence is unreliable and not useful for the goal of determining adult age for several reasons." I would disagree, as the 2019 study by Gambier et al (Contribution of third molar eruption to the estimation of the forensic age of living individuals) comes to different conclusions. Please include them in your discussion.

We added a paragraph in the Discussion section about this study.

Reviewer 2 Report

The paper presents important contribution in applying Demirjian method to estimate legal age in South Italy population. However, I suggest that, prior to publication, authors consider following issues and modify the manuscript accordingly:

-        The first part of the introduction lacks references on relevant methods and studies that are listed there.

-        From introduction section, it is not clear why authors are conducting the study, are methods generally population specific, what has been done on Italian population samples so far? This is also important to mention in Discussion. Can results be generalized to the Italian population? Why Cameriere’s method was not mentioned?

-        Authors are first mentioning the modified Demirjian method [7,1] in introduction and then in M&M, if I understood well, their modification of Demirjian method. In any case the general method, and its modifications should be explained in more detail, especially considering the multidisciplinary of the journal and the special issue. Besides, it is not clear, for example, why lower molars are separately evaluated.

-        Obtained χ2 and P-values for between sex comparison and sides should be reported. P>0.05 is not adequate in this case.

-        Order of results in table should be updated in more logical manner, i.e., last three rows should contain results of lower molars.

-        The main results should be reported in more detail. I suggest that authors create table that will include threshold, accuracy, sensitivity, specificity, PPV, and NPV.

-        Discussion is generally not written well. The first two sentences with the main findings are OK, but text from ln. 142 to 171 is more appropriate for introduction then for discussion. The authors should discuss previous works in the light of their findings and focus on implications of their study.

-        I suggest that the authors remove the parenthesis from the title and include important information in the title itself.

-        Authors should provide information on ethical approval

minor issues

-        age estimation does not need to be capitalized

-        please use estimate instead of to determine

-        please change gender to sex

Author Response

The paper presents important contribution in applying Demirjian method to estimate legal age in South Italy population. However, I suggest that, prior to publication, authors consider following issues and modify the manuscript accordingly:

Thank you very much for your appreciation and for the comments and suggestions.

-        The first part of the introduction lacks references on relevant methods and studies that are listed there.

We tried to improve the Introduction with more information and more references.

-        From introduction section, it is not clear why authors are conducting the study, are methods generally population specific, what has been done on Italian population samples so far? This is also important to mention in Discussion. Can results be generalized to the Italian population? Why Cameriere’s method was not mentioned?

We tried to improve the section according to your suggestions.

-        Authors are first mentioning the modified Demirjian method [7,1] in introduction and then in M&M, if I understood well, their modification of Demirjian method. In any case the general method, and its modifications should be explained in more detail, especially considering the multidisciplinary of the journal and the special issue. Besides, it is not clear, for example, why lower molars are separately evaluated.

We tried to explain better in the text the differences between the methods.

-        Obtained χ2 and P-values for between sex comparison and sides should be reported. P>0.05 is not adequate in this case.

We reported the χ2 and P-values for between sex comparison and sides.

-        Order of results in table should be updated in more logical manner, i.e., last three rows should contain results of lower molars.

We changed the table as requested.

-        The main results should be reported in more detail. I suggest that authors create table that will include threshold, accuracy, sensitivity, specificity, PPV, and NPV.

We created another table with the requested values.

-        Discussion is generally not written well. The first two sentences with the main findings are OK, but text from ln. 142 to 171 is more appropriate for introduction then for discussion. The authors should discuss previous works in the light of their findings and focus on implications of their study.

We moved a part of the Discussion in the Introduction section and try to improve the Discussion section with more comments and comparisons with our results.

-        I suggest that the authors remove the parenthesis from the title and include important information in the title itself.

We changed the title according the instruction of both reviewers.

-        Authors should provide information on ethical approval

We did not request the documentation since no new therapies or new radiographic investigations were carried out specifically for this study, but we used data already present in the medical records of previous treatments (treatments that also cover very long periods of time, both because of the treatment times specific to the individual patients and because of the waiting lists).

minor issues

-        age estimation does not need to be capitalized

-        please use estimate instead of to determine

-        please change gender to sex

We did these modifications.

Round 2

Reviewer 1 Report

I appreciate the revised article, and the authors have taken almost all comments into account. However, there are still some minor recommendations that should be revised and incorporated.

- The inserted Table 2 should be referred to in the text.

Author Response

I appreciate the revised article, and the authors have taken almost all comments into account. However, there are still some minor recommendations that should be revised and incorporated.

- The inserted Table 2 should be referred to in the text.

Thank you for the appreciation. We added a reference of the table 2 in the text.

Reviewer 2 Report

Authors have considered most changes and substantially improved the article. However, in my opinion, there are still several issues that should be considered prior to the publication of the paper:

-       Discussion has been improved but it is still difficult to follow, and findings are not considered in logical order. Authors mention “many studies” that validated such approach, but it is not visible in the Discussion, where the authors report the relevant parameters only from one study. It is important to report that some proportion of individuals had molars in some stage, but according to the title and the aim of the paper, authors should focus more to the classification performance of their approach. Except for discussing findings, authors should pay more attention to text style and the flow of sentences.

-        Population specificity is not mentioned again in introduction, is method population specific or not? This should be one of the main arguments why the study was conducted in the population.

-        For the sentence “The gold standard for the assessment of bone growth is the hand and wrist X-ray.” author cite reference [6] -which does not mention hand and wrist at all. I assume the order of references changed at some point, but authors really need now to double-check everything.

-        p 3 ln 100. This age analysis is….. -> This sounds strange. Please modify to include “method”

-        p 3 ln 102 In the literature there are many studies; p 5 ln 182 “studies have shown a ….”-> Which studies? This statement requires references.

-        p 3 ln 122 Age Estimation is still capitalized. But, were they ask to score the degree of development or estimate age, please be précised.

-        Table 2 is not mentioned in the text.

-        Table 2 – “All the values are expressed as percentages. “ is not necessary because authors included % in the table.

-        Please separate main findings in the discussion from the rest of the discussion with a new paragraph. In my opinion, main findings are ln 176-182

-        p5 ln 183. “Different methods for the dental assessment of the patient's age have been” Again, more introduction-like sentences. Please start from YOUR findings and compare it to the other relevant studied and explain what they mean.

-        p 6 ln 220 - Findings are compared, but the order should be adjusted. Please, first discuss inter/intraobserver consistency, then, compare it to other studies. After that, male and female, etc.

-        p 6 ln 227 age range is listed in squared brackets like references

-        p 6 ln 234 Anyway, these findings were consistent with previous studies – One study is referenced?

-        If the population is important in such cases, please mention that study conducted on particular population provided the result.

-        p 6 254-263 Authors suggest larger population studies, but they did not consider that samples should be also balanced according to regions. In their case, samples originated from the south of Italy, so it is possible that it could not “capture” whole variability of Italian population.

Additional consideration.

-        I apologise for not mentioning this at the first round, but I suggest that, since we are dealing here with classification and not prediction issue, after mean ages authors write how many participants were younger and older than legal age. This way readers will know categories are balanced and if results could be affected by the prevalence.
